# Collaborative Governance for Participatory Regeneration Practices in Old Residential Communities within the Chinese Context: Cases from Beijing

**Zixuan Zhang** **, Junchen Pan and Yun Qian ***

School of Landscape Architecture, Beijing Forestry University, Beijing 100083, China;
zixuan7@bjfu.edu.cn (Z.Z.); panjunchen@bjfu.edu.cn (J.P.)
* Correspondence: qianyun@bjfu.edu.cn

**Abstract:** The regeneration of old residential communities in China is one of the most important tasks in urban renewal. In recent years, distinctive models and pathways have emerged in the emerging practice of participatory community regeneration, all of which can be seen as applications of collaborative governance theory at the community level. Collaborative governance is considered an effective way to achieve multiple goals in urban regeneration, but there has been relatively little research on collaborative governance in small-scale regeneration projects. This paper summarizes three nuanced pathways in the collaborative governance model through case studies, which are led by different initiators, resulting in collaborative models, rights dynamics, and implementation pathways that are applicable to different scenarios, effectively resolving community conflicts and producing sustainable practical results. This study compares how the three models achieve their respective objectives in participatory regeneration projects by coordinating the different stakeholder participation processes. These three models complement and extend international experience and will provide a vivid Chinese example for other developing countries around the world.

**Keywords:** collaborative governance; participatory regeneration; regeneration modes; stakeholders; Chinese context; old communities



## 1. Introduction

China's urban construction has entered the stage of stock space improvement and efficiency. As of 2020, there were about 170 thousand old residential areas in China, covering hundreds of millions of people[1]. Due to factors such as complex social structures [1], unclear management responsibilities, and limited spatial development [2], old communities have experienced varying degrees of decline in physical space and community cohesion. The regeneration of old communities has become an important affair of urban regeneration and social governance in China. Considering the two key factors—government intervention and the level of citizen–private–social participation [3], the three main governance models of urban regeneration in China are state-led, market-led, and collaborative. There has been a great deal of research and practice on state-led and market-led urban regeneration, and one thing they have in common is that they tend to focus on large-scale regeneration projects, such as reshaping the urban landscape, reviving historic districts, etc. In contrast, the regeneration of residential communities is extremely small in scale but large in scope, involving many sectors and complex relationships between stakeholders. With the trend of new urbanization shifting from a focus on land resources to citizens [4,5], there has been a promotion of bottom-up citizen awareness [6,7]. Many planners and scholars believe that top-down methods may no longer be applicable to handling complex community conflicts [8,9]. Collaborative governance focuses on coordinating and involving different public, private, and civil society stakeholders [10,11], emphasizing consensus building, col-

lective decision making, and achieving common goals [12,13], and is, therefore, considered an effective model in community regeneration.

Collaboration has been recognized as an effective tool for community planning and governance in Europe [14], the United States [15], Japan [16], and Taiwan, China [17]. Policymakers increase public trust and enhance social capital by interacting, collaborating, and sharing knowledge and perspectives with stakeholders, thereby improving the quality and rationality of planning decisions [18]. Scholars have conducted extensive research on the modes [19], mechanisms [20], toolkits [21], rights dynamics [22], and effectiveness evaluation of collaborative planning [23]. In terms of models, most of these projects are initiated by various social organizations (non-profit organizations) and are usually implemented in collaboration with social organizations, grassroots governments, local community organizations, and local businesses [22,24,25]. The government largely authorizes various social institutions, enabling them to form consultation platforms with professional social workers or volunteers to provide services such as public opinion collection, construction and maintenance technical support, implementation process coordination, and local maintenance and management [26,27]. In terms of approaches, in order to achieve better public participation, projects often involve multiple stages, including continuous public science training, non-institutionalized communication and collaboration among participants [28], and the application of digital participation platforms and media tools [29], with durations ranging from months to years [30].

In recent years, many cities in China have been engaging in participatory regeneration practices for old communities through small-scale space co-creation and collaboration among multiple stakeholders. These projects have achieved significant effects in improving the community environment [31], strengthening neighborhood networks [32–34], enhancing public participation [35,36], and accumulating social capital [37]. Cases from Shanghai emphasize the significance of social organization participation in constructing platforms for multiple stakeholders [38,39]. Studies from Guangzhou explore the co-governance of communities and enterprises [40,41]. Cases from Beijing confirm the effectiveness of CRP (community responsibility planner) systems [42,43]. These "top-down" and "bottom-up" power dynamic models demonstrate the achievements of different social entities in innovative social governance practices, despite their different backgrounds, compositions, and social relationships. However, due to the special national conditions of China where the development of social organizations and public participation systems is still in its early stages [44], the mechanisms and methods of these collaborative governances mostly draw on the experiences of participatory planning internationally. Although various community practices that have emerged across China show more effective or Western-like models, actual practices differ significantly from the regular procedures [45]. First, China's public participation model is still government-led and citizen-passive [46]. Grassroots governments tend to play a supporting and supervisory role throughout the entire process, rather than fully empowering other stakeholders [47]. Second, the development of various social organizations in China is relatively weak. Long-term top-down policy formulation and implementation have to some extent inhibited the initiative of social service enterprises and social organizations [48,49].

However, the collaborative mechanisms among multiple stakeholders in participatory regeneration are complex, especially in the Chinese context mentioned above, as it is not easy to manage conflicts between the many stakeholders. As a result, especially for older community small-scale regeneration projects, there has been less research into collaborative governance models. The specific implementation pathways and cooperative interactions between the different stakeholders in this type of practice are not clear in the previous literature.

To address these questions in the literature, we studied six cases of old residential community regeneration. All of these cases used collaborative governance models. Specifically, we propose three questions that have not yet been fully addressed in studies of collaborative governance:

(1) What are the modes and pathways of collaborative governance in participatory community regeneration practices in the Chinese context?

(2) When compared horizontally, what are the similarities and differences among these modes? Which scenarios are they respectively applicable to?

(3) What is the relationship between the different stakeholders within the collaborative governance processes?

Exploring the answers to these three questions has important implications for researchers and grassroots administrators. Understanding the specific collaborative governance mechanisms and pathways at the community level can help grassroots governments to fine-tune their community regeneration projects and is important for exploring the process of innovative social governance.

This paper is structured as follows. First, we present the theoretical background of the study, highlighting the important role of collaborative governance theory in participatory community regeneration. We then present representative cases of participatory community regeneration in Beijing in recent years, summarizing and comparing the similarities and differences between the modes to which these cases belong in order to clarify how collaborative governance models can play an effective role in participatory community regeneration.

## 2. Materials and Methods

We take Beijing as a case study city and analyze 6 recent cases of old residential communities in central urban areas (Figure 1) that reflect different mainstream modes and patterns of current participatory regeneration for old residential communities in China. Data and information have been collected and interpreted primarily in the framework of grounded theory. The basic situation for the six cases derived from fieldwork, relevant policy documents, and semi-structured interviews can be seen in Table 1. All six cases were initiated between 2018 and 2019 and have continued to have social effects after completion.

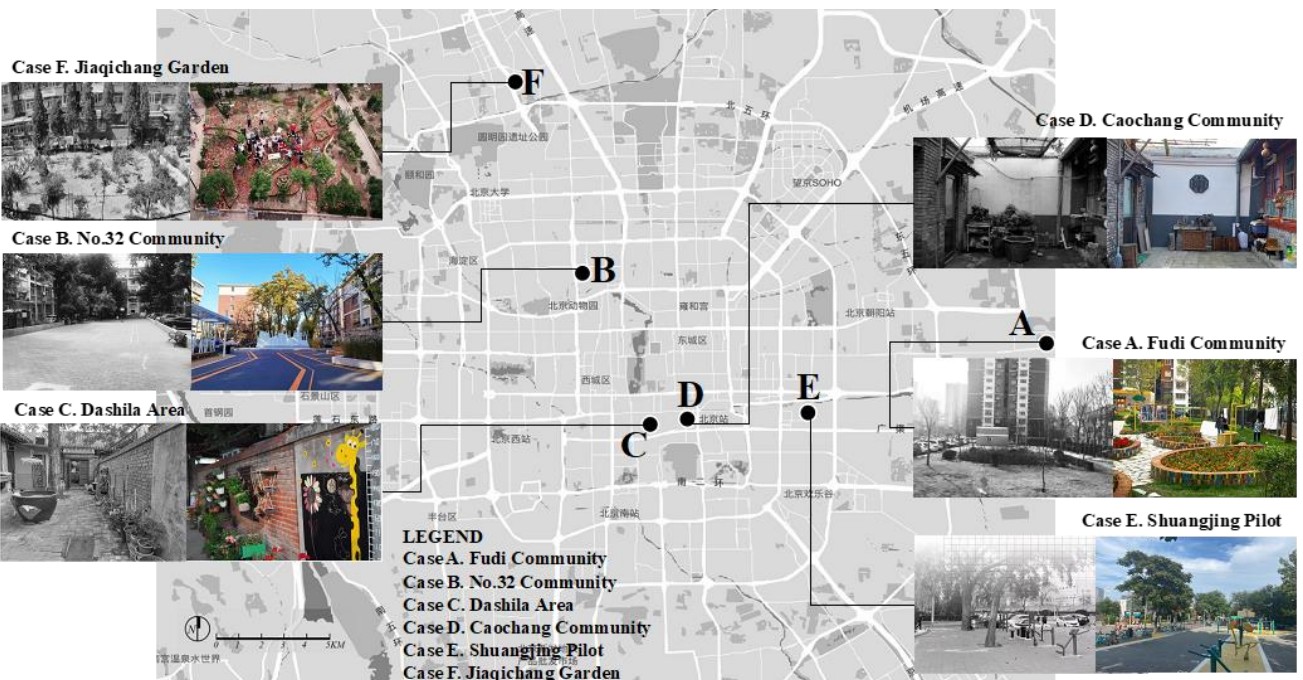

**Figure 1.** Location of the six cases in Beijing. (Figure source: made by the author).

**Table 1.** Basic statistics for the six cases.

| | Name of Project (Chinese) | Main Units | Regeneration Area | Street Office | District | Scale of Regeneration Area |
|---|---|---|---|---|---|---|
| Case. A | "Rose fairy tale garden" (玫瑰童话花园) | CCRP team; XVII Studio | Fudi Community | Changing Street | Chaoyang District | 1300 m$^2$ |
| Case. B | "Our land" (咱家这块地) | Responsibility planner team for North Taipingzhuang Street; BJFU team | No. 32 Community | North Taipingzhuang Street | Haidian District | 1200 m$^2$ |
| Case. C | "Planning of Dream Garden." (梦想花园计划) | Dashila Street office; DIC; BJFU team | Dashila Area | Dashila Street | Xicheng District | 4 alleys |
| Case. D | "Zero waste recycling courtyard." (零废循环小院计划) | SSWS; BJFU team | Caochang Community | Qianmen Street | Dongcheng District | 1 alley |
| Case. E | "Well No.1" (井点一号) | Responsibility planner team for Shuangjing Street; UTC; SDS | Shuangjing Pilot | Shuangjing Street | Chaoyang District | 300 m$^2$ |
| Case. F | "Garden of happiness" (幸福花园) | Responsibility planner team for Qinghe Street; SNS | Jiaqichang Garden | Qinghe Street | Haidian District | 270 m$^2$ |

Abbreviations of entities are used in the table: CCRP (Changying Community Responsibility Planner), BJFU (Beijing Forestry University), DIC (Dashila Investment Company, Beijing, China), SSWS (Seeding Nature Studio), UTC (Seeding Nature Studio), SDS (Seeding Nature Studio), SNS (Seeding Nature Studio).

By identifying the background, initiator, and initiation process of each case, they were grouped into three recognizable modes: the Planning Devolution of Governments (PDG) mode; the Practitioner's Innovation Leadership (PIL) mode; and the Urban Regeneration Collaborative Partnership (URCP) mode (the characteristics of each mode are described in detail in Section 3). The premise of the article is that these initiatives are largely distinct from regular government-led, one-dimensional, one-time renovation projects, even if their successful operation relies heavily on government support. On the one hand, they are interdisciplinary (mainly from the fields of planning sciences and social sciences) integrated teams that can complement each other's knowledge shortages and, secondly, they connect top-down and bottom-up.

Case A. Regeneration of green space in the Fudi Community. The Fudi Community was built in 2008 and is located in Changying Town Street, Chaoyang District, Beijing. The Fudi Community was built in 2008. Due to the irrational space planning and facility layout, a large number of elderly people and children in the community exercise on narrow building paths or non-motorized aisles, resulting in the service potential of the site not being released. The project involved the renewal of a residential green space of approximately 1300 square meters within the Fudi Community. In 2019, the ZhongShe Social Work Development Foundation (ZSWDF) launched the "Micro Space Sunshine Rebirth—Chaoyang District Small Micro Public Space Regeneration Program." The site was selected through an open proposal process, and the XVII Studio (affiliated with the School of Architecture of the Central Academy of Fine Arts) and Changying community responsibility planner (CCRP) team were chosen to collaborate on the project, resulting in the construction of a "Rose Fairy Tale Garden".

Case B. Public space renewal of No. 32 Community. The No. 32 Community on Xueyuan South Road was built between the 1950s and 1980s and consists of 2350 households. The volume ratio of the area is very high, and the community is seri-

ously aged. The project focused on an area of approximately 1200 square meters, enclosed by a short concrete wall and lacking appropriate resting facilities and public activity spaces. In 2020, the community responsibility planner team from North Taipingzhuang Street and their partners from universities in Haidian District launched the "Building North Tai Together, Our Home" project to update the public spaces in the No. 32 Community.

Case C. Sustainable regeneration of the Dashila Area. The project is located in the traditional Dashila Hutong community of Xicheng District, Beijing, where most residents have long lived in crowded and cluttered spaces formed by the continuous construction of old courtyard houses built in the 1600s. The need for small green spaces is strong. Since 2018, a joint research and creation team of professionals and students from Beijing Forestry University's urban planning, landscape architecture, and ornamental horticulture departments, together with Dashila Investment Company (DIC), Beijing, China, a real estate and commercial services company, have been working on the "Dream Garden Plan" project with the support of Dashila Street Office.

Case D. Green-renewal practice in the Caochang Community. The Caochang community is a traditional Hutong community on Qianmen Street, Dongcheng District, Beijing. With the support of Qianmen Street and the Caochang Community, as well as funding from the Vanke Public Welfare Foundation, the Sanzheng Social Work Studio (SSWS) and Beijing Forestry University (BJFU) teams jointly launched a public welfare project called "Green Micro Renewal Plan—Zero Waste Recycling Small Courtyard".

Case E. Smart community governance of Shuangjing Pilot. Shuangjing Street, located in the central–western part of Chaoyang District, Beijing, has a large population, a complex environment, a dense road network, and developed businesses. The site is located at the corner of the street between Jiulong Community and Beijing Station, an unused high platform of about 300 square meters. In 2019, Beijing Urbanxyz Technology Company (UTC) (Beijing, China), acting as a community responsibility planner and collaborating with Sketchaction Design Studio (SDS), used technological and social innovation to promote the "Intelligent Governance" practice in the Shuangjing Pilot program.

Case F. Green building of Jiaqichang Garden. Jiaqichang Garden is located in the Meiyuan Community, Qinghe Street, Haidian District, Beijing, and is a typical aging community with 384 households built in 1967. In 2019, relying on the "Qinghe Experiment" project at Tsinghua University, Seeding Nature Studio (SNS) also acted as a community responsibility planner, working with local social work organizations and others to create a "Happiness Garden".

## 3. Modes of Participatory Regeneration in Chinese Residential Communities

Case A (Fudi Community) and B (No. 32 Community) represent the PDG mode, case C (Dashila Area) and D (Caochang Community) represent the PIL mode, and case E (Shuangjing Pilot) and F (Jiaqichang Garden) represent the URCP mode.

### 3.1. The PDG Mode

The biggest feature and advantage of this mode is the introduction of third-party planning experts from different fields and departments, which serve as bridges to effectively connect various stakeholders and departments. Third-party planners assume a variety of different roles such as organizers, coordinators, and implementers. Thus, the PDG is a typical multi-level governance mode that provides opportunities for dialogue, consultation, and cooperation between different stakeholders and promotes trust and transparency in decision making.

3.1.1. The Third-Party Planning Experts—To Bridge the Connections between Stakeholders and Government

"Community planner" first emerged in the 1960s as part of the exploration outcomes of humanistic and multi-objective approaches [50]. In the 1970s, community planning became a major method of urban renewal in Europe and the United States [51]. In main-

land China, community planning has developed based on nearly a decade of experience in community renewal and governance. Since 2010, cities such as Shanghai, Shenzhen, and Chengdu have borrowed from the experiences of Japan and Taiwan and attempted to establish a system of community planners. In 2017, Beijing introduced the community responsibility planning (CRP) system, which was then promoted citywide in 2019. The district government hires full-time or part-time professional planners to provide technical services for street planning, construction, and operation, guide public participation, and bridge the gap between the government and residents [52]. Subsequently, policy documents were issued by various districts of Beijing to provide specific guidelines for implementing the CRP system. For example, the "Sunflower Seeds" project in Chaoyang District consists of 36 long-term incumbent responsibility planners and one short-term hired planner. Most planners in most districts are usually hired on a part-time basis for one year, ensuring professionalism and flexibility in the implementation process.

The PDG mode, as an extension and devolution of the government's street-level administrative organization in the planning field, plays a role similar to a boundary spanner (a person who works between public and private stakeholders and bridges the gap between them) [53,54]. On the one hand, the CRP system follows the vertical relationship between the Beijing municipal government, district government, and street, which ensures top-down policy dissemination. On the other hand, the CRPs (community responsibility planners) also empower intervention as a strategy to overcome the problem of insufficient participation of marginalized stakeholders, such as residents and social organizations in community governance. The implementation pathway of the PDG mode consists of four stages:

(1) Stage 1: Platform Construction and Participation Expanding.

The establishment of a collaborative platform is the first step in multi-party participation in the PDG mode [55]. Typically, the PDG mode connects with pilot projects of the district government, which usually have characteristics such as a large population, a large site area, and complex usage scenarios. Therefore, CRPs need to establish a cohesive collaborative structure with neighborhood committees (grassroots governments), residents (community agents), property management, social organizations, and design teams. In the case of the No. 32 Community, CRPs established a promotion team in the early stage to discuss planning procedures, policies, and expected results with residents, and spread the core concept of "co-building and co-governance".

(2) Stage 2: Co-Design and Decision Making.

Guiding collaboration and reaching consensus are the core processes of participatory regeneration in the PDG mode [56]. Community planners use multiple public participation activities to guide residents to propose needs, and then accurately translate these needs into planning and design language to help decision making. The forms of participation usually include participatory seminars and design workshops. For example, in the Fudi Community, the CCRP team encouraged residents to express their design concepts through group discussions and presentations. Then, the XVII Studio conducted multiple rounds of adjustments through an expression–response process (Figure 2). Later, the community planners also invited experts from various fields such as planning, design, and social work for on-site reviews, and organized online selection activities to increase opportunities for communication between residents and professionals. In the case of the No. 32 Community, the CRP team used methods, such as selecting preferred maps and making collages, to guide residents' participation (Figure 3), and finally jointly decided to transform the site into a community activity and meeting square, which can accommodate activities, neighborhood interactions, children's fitness, and elderly chess playing.

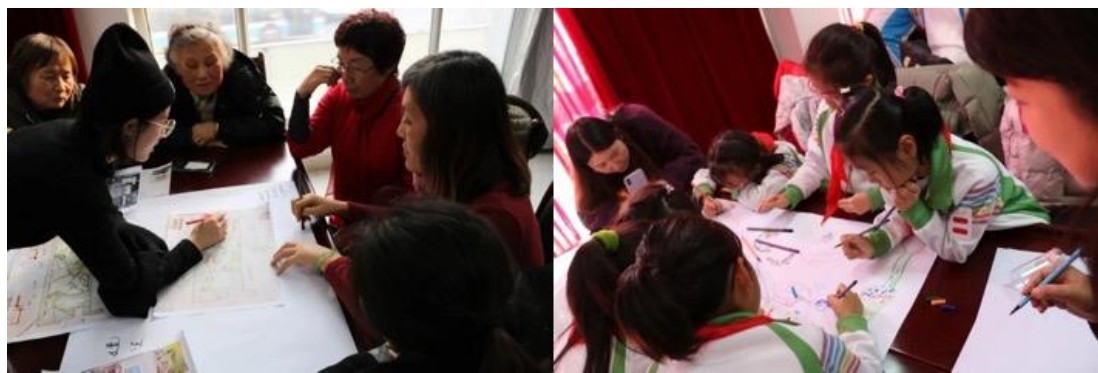

**Figure 2.** Participatory design workshop in the Fudi Community. (Figure source: Beijing Capital Engineering Architectural Design Co., Ltd., Beijing, China, WeChat platform, 2019).

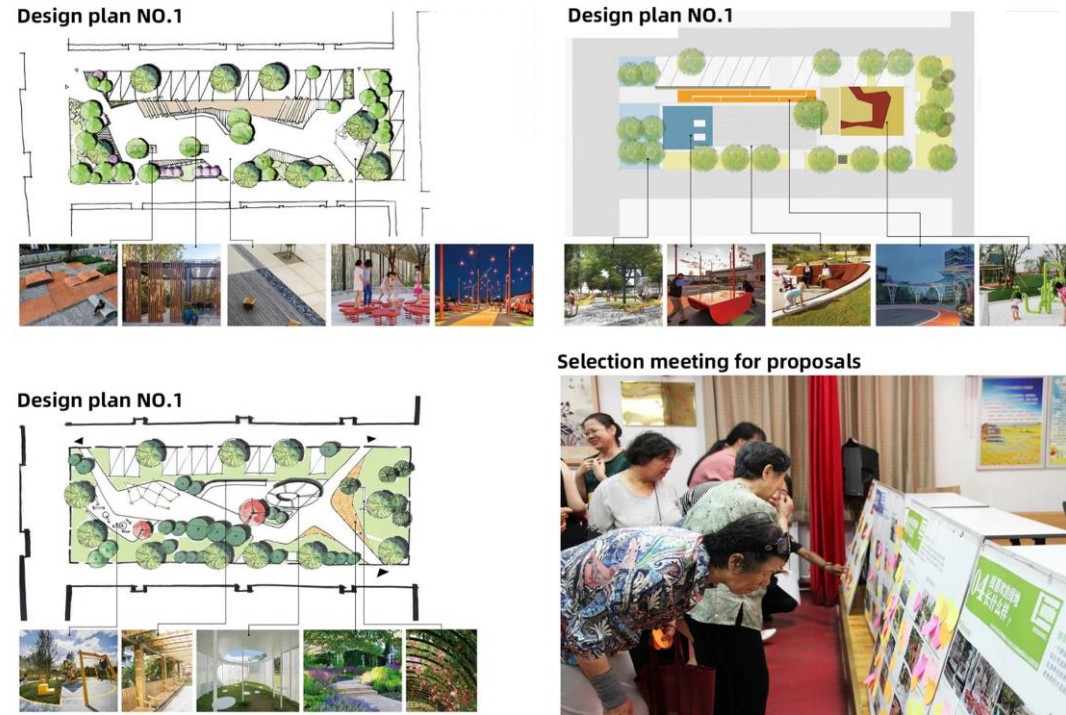

**Figure 3.** Participatory design workshop in the No. 32 Community. (Figure source: Beijing Municipal Commission of Planning and Natural National Resources Commission WeChat platform, 2020).

(3)    Stage 3: Co-construction.

Public participation in the creation process is challenging due to the large number of stakeholders. The CRPs need to coordinate different issues that construction units, communities, residents, and volunteers run into during construction, in addition to providing guidance for implementation. The regeneration area of the Fudi Community is relatively large, involving multiple functional zones and construction processes. The main hard decoration was finished by the construction team, and about 50 locals actively took part in the landscape wall graffiti, flower bed masonry, sign making, and flower planting. In the case of the No. 32 Community, the CRPs oversaw not only the building and landing of structures like galleries, stages, tables, and chairs but also the ongoing modification of the plant's landscape and nighttime lighting plans (Figure 4). This places higher requirements on the refined governance and overall management capabilities of the CRPs.

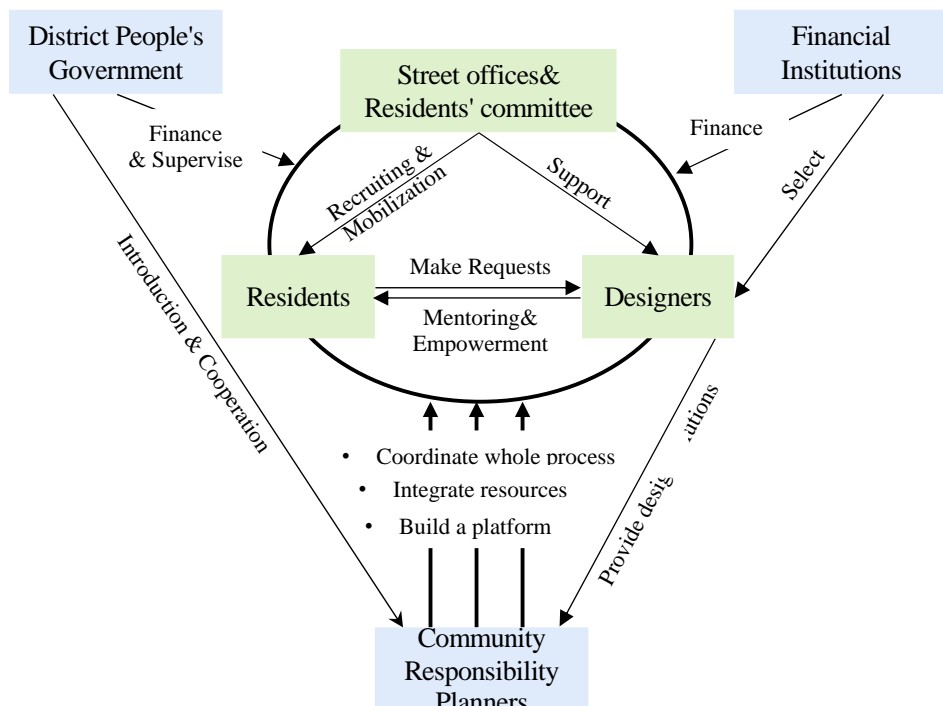

**Figure 4.** Relationship of the PDG mode with stakeholders. (Figure source: made by the author).

(4)　Stage 4: Spatial Activation and Long-Term Maintenance.

The CRPs promote neighborhood interaction and stimulate community vitality by enlivening the public spaces through cultural promotion, art exhibits, and other cultural activities. In order to ensure the regeneration results are sustainable, CRPs typically establish a community joint arrangement mechanism to guide residents to consult and manage maintenance modes and track the use and maintenance of the site for a long time. Since the Fudi Community Garden's completion in September 2020, four public welfare events have been held, including a seed program, garbage classification, plant printing, co-painting, etc. Additionally, the site has served as a natural learning ground for the Sun District's second experimental primary Fudi school, generating many positive social influences.

Table 2 summarizes the specific processes in the four stages in the two cases of the PDG model and the approaches they took to public participation.

### 3.1.2. As Organizers, Coordinators, and Implementers

The PDG is a typical multi-level governance mode that provides opportunities for dialogue, consultation, and cooperation between different stakeholders and promotes trust and transparency in decision making [18]. As an organizer, the CRP has built a solid collaborative structure, including a top-down policy communication chain and a bottom-up communication feedback mechanism, which enhances residents' sense of community belonging, and responsibility. As a coordinator, the CRPs coordinate the allocation of powers among stakeholders. As a co-builder, CRPs integrate human and material resources to continuously expand the depth and breadth of public involvement. Overall, this is a low-intervention model that plays an important role in ensuring the dynamic balance of the parties involved.

Figure 3 summarizes how CRPs construct a network of stakeholders to form a stable model of cooperative community regeneration based on shared participation and decision making in the PDG mode.

**Table 2.** Process and participation methods in the PDG mode.

| Stage | Specific Process | Participation Methods | |
| --- | --- | --- | --- |
| | | Fudi Community | No. 32 Community |
| Platform Construction and Participation Expanding | Connecting resources | Street declaration of pilot projects, building platforms with teams of CCRP team, XVII Studio, volunteer groups, etc. | North Taipingzhuang Street government, planners, and university partners' cooperation |
| | Spreading the idea | Planners and university partners hold community meetings | |
| Co-Design and Decision Making | Pre-stage research and analysis | Collecting residents' needs through surveys and questionnaires | The university partners team conducted a detailed study of the site |
| | Leading co-design | The CCRP team, XVII Studio, and the community council held a participatory workshop | Guiding residents to design their gardens in the form of puzzles |
| | Preliminary plan | The XVII Studio helps residents draw plans | The university team distilled the residents' ideas into three preliminary design proposals |
| Co-construction | Further adjustment | Invite experts to review the program, followed by online selection | Consider residents' opinions, hold a public vote |
| | Reaching consensus | Multiple rounds of discussion to determine the final plan | Determine the final function, orientation, and activity facilities |
| | Implementation | Residents, neighborhood councils, and volunteers participate in co-construction | The planner coordinates the construction unit for implementation |
| Spatial Activation and Long-Term Maintenance | Post-maintenance and operation | Community charity activities, such as seed projects and plant printing | The site is maintained by the property |
| | Post-evaluation | Regular visits by CRPs | |

### 3.2. The PIL Mode

The biggest feature and advantage of the PIL mode is the introduction of innovation institutions instead of just the third-party planners in the PDG mode. The innovation institutions could exercise their expertise in community practice, including planning, finance, and technology. As seen from the name, the dominant characteristic of the PIL mode is innovation, in all fields.

3.2.1. The Collaborative Partnership of Innovation Institutions—To Exercising Expertise in Community Practice

The PIL mode represents a cross-sector partnership consisting of innovation institutions or enterprises comprising professional individuals or teams, including but not limited to local universities, research institutes, and social enterprises oriented toward interests or social benefits. Most of them are professional practitioners in the field of urban planning or social work. International scholars have explored the crucial function of universities as institutions that gather a range of social resources and knowledge technologies [57], contributing uniquely to fields such as urban renewal and smart cities [58]. Similarly, they have also studied how community-based social enterprises achieve community renewal by providing public participation services and facilities for residents [59]. In recent years, many Chinese scholars have established analytical frameworks for various innovative organizations led by social innovation and social governance perspectives [60,61]. These studies and practices have demonstrated that professionals have many advantages in providing innovative community regeneration methods and expanding related knowledge.

In the PIL mode, these practitioners are characterized by a small scale and high flexibility, and they emphasize a bottom-up approach, creativity, and long-term benefits. The practice includes three main stages:

(1)　Stage 1: Customized Regeneration Planning.

These practitioners or research and innovation institutions often prioritize the regeneration of small-scale public spaces. After conducting scientific and detailed site investigations, they provide tailor-made planning schemes according to local conditions (Figure 5), which is their advantage. For example, in the Dashila case, the BJFU team customized renovation plans for four courtyards based on their unique needs and architectural characteristics. Targeted and distinctive overall planning provides the basic principles and ideas for the implementation of subsequent projects and a series of public participation activities.

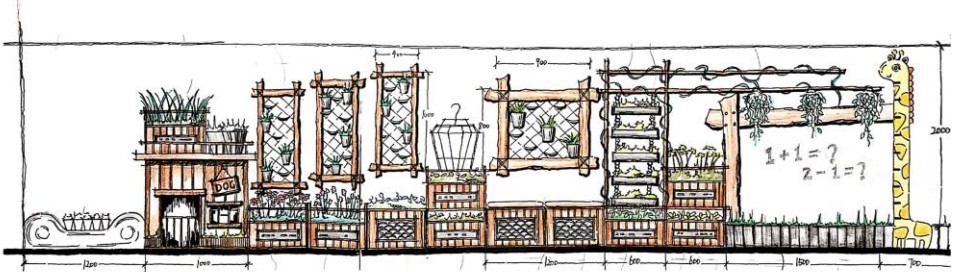

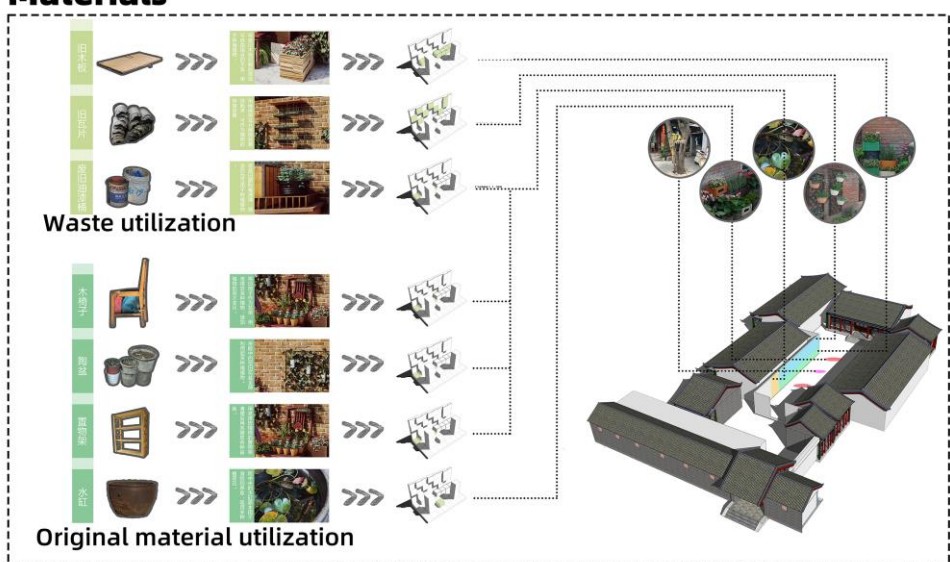

**Figure 5.** Design solutions. (Figure source: green regeneration WeChat platform, 2019).

(2)　Stage 2: Co-building of Scenes.

Real-time interaction and feedback during the design and construction process put a lot of pressure on practitioners' ability to provide services for diverse and dynamic needs. In the Dashila case, the university team provided precise and deductive design concepts and strategies. After fully soliciting residents' opinions, volunteers from the school, neighborhood committee, and residents jointly built flower beds, planted vegetation, and created multiple green scenes with different themes (Figure 6). In this process, the BJFU team communicated fully with the residents and realized their ideas. They also modified the construction plan in real-time based on residents' feedback, making the results of the construction most suitable for residents' usage scenarios.

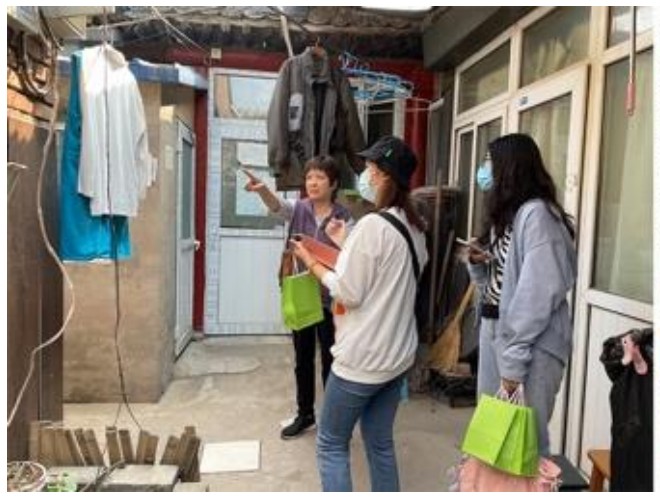
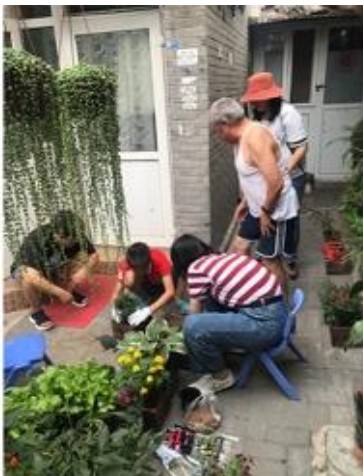
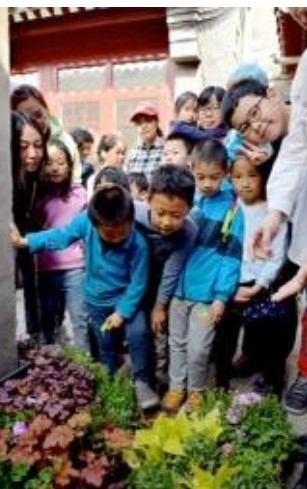

**Figure 6.** Photos of co-building. (Figure source: Beijing Practice Group of Nostalgia of Beijing Forestry University, 2019).

(3)  Stage 3: Maintenance and Operation.

Due to the full co-building process and good spatial transformation effect, daily maintenance is generally completed independently by residents. In the initial stage of the project, the team usually conducts online and offline visits every week and invites experts to provide intellectual and technical support. In the Dashila case, the university team invited experts in horticulture to use the constructed green scenes to conduct green education courses such as plant cognition and flower cultivation for children, guiding residents to gradually develop the habit of green construction. In the Caochang case, the SSWS fully displayed the results of residents' use of the space by continuously inviting multiple media outlets to report on it.

Table 3 summarizes the specific processes in the three stages in the two cases in the PIL model.

**Table 3.** Process and participation methods in the PIL mode.

| Stage | Specific Process | Participation Methods | |
|---|---|---|---|
| | | **Dashila Area** | **Caochang Community** |
| Customized Regeneration Planning | Mobilization and recruitment | Holding community mobilization meetings | |
| | Summary of residents' needs | In-depth interviews to identify needs | |
| | Program generation | Propose several design sketches and solicit residents' opinions | |
| Co-building of Scenes | Participatory construction activities | The BJFU team and residents build planters and grow plants together | Social workers, the BJFU team, and building with residents |
| Maintenance and Operation | Studying and training activities | University experts hold activities, such as plant awareness for children | University students hold participatory workshop activities |
| | Daily Maintenance | Residents maintain by themselves | |
| | Operation activities | Series of exhibitions, communications, and visits | Series reports and communications |

3.2.2. Specific Roles

As an innovator. Community-based revitalization is a complex project involving a wide range of fields, diverse services, product types, and processes ranging from 1–2 years to 3–4 years. Innovative organizations can derive a series of derivative services, such as academic salons, public welfare activities, handmade workshops, and cultural and creative

products, and space achievements can serve as natural education bases in the community. These achievements can be seen in examples, such as the "Dream Garden Plan" theme exhibition and the "Old Items, New Green" gardening workshop in Dashila and Caochang.

As an enabler. The practitioners can use academic achievements and social resources of universities for interdisciplinary social innovation, providing technical support and empowerment in multiple fields such as urban research, landscape design, plant construction, information technology, and social work. For example, in the Caochang project, the BJFU team designed courtyard landscape schemes and planned workshop activities based on the theme of "zero waste". Community social work organizations help disseminate relevant concepts and establish community conventions.

Figure 7 shows a simple and efficient communication process among the stakeholders in the PIL mode.

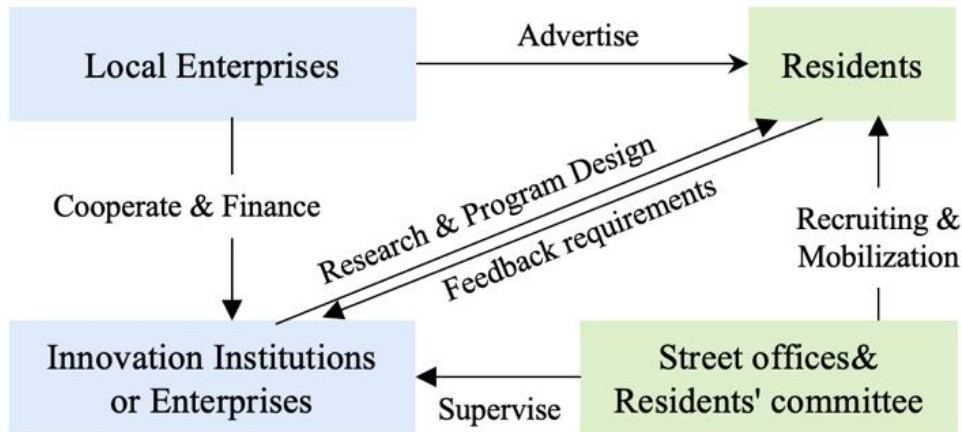

**Figure 7.** Relationship of the PIL mode with stakeholders. (Figure source: made by the author).

### 3.3. The URCP Mode

The biggest feature and advantage of this URCP mode is the introduction of a wide range of market forces, which aim to develop a commercially viable mode. The companies of the URCP often play the role of intermediaries. The involvement of market forces is the fundamental factor that distinguishes this mode from the other two modes.

#### 3.3.1. Cooperation of a Wide Range of Market Forces—To Develop a Commercially Viable Model

As mentioned earlier, the traditional regimes of renovating old residential neighborhoods have to some extent hindered the participation of market forces. However, in recent years, more and more enterprises have joined the field of community construction [62]. The driving force of the URCP mode often comes from the cooperation between the government and enterprises [49] or the cooperation between university institutions and enterprises (i.e., profit) [37]. Although there is some similarity in the cross-departmental cooperation approach with the PIL mode, the essential difference between the URCP and the PIL is the fact that this cooperation is based on production and consumption [63]. They possess the investment capital and product development capabilities of the company, the trust of the district government and residents, the professional skills of planners, and social responsibility.

In the two URCP mode cases in this paper, the Pilot on Shuangjing Street was initiated by a local technology company, whose head was also a part-time community responsibility planner in Chaoyang District; and the Jiaqichang Community Garden construction was undertaken by the SNS, a team incubated by Tsinghua-THUPDI and a CRP team on Qinghe Street, Haidian District. It can be seen that the URCP is a model of collaboration on a larger team scale, encompassing many active participants with complex identities who are concerned with community governance. These collaborative partnerships accumulate social

capital through the complex network of interactions among multiple stakeholders, which helps to establish a common identity and goal, more effectively disseminate information, and promote consensus and action [64,65], and this social capital in turn encourages urban regeneration operators (URO) to expand product types, improve service quality, and reduce operating costs [66]. In summary, it is a business-focused, government-facilitated, commercial model that can be operated sustainably by increasing social capital. The pathway of this model consists of four stages:

(1)　Stage 1: Research and Preparation.

In order to achieve significant results, the URCP tends to choose public spaces with a wide range of users, unresolved social conflicts, and controllable difficulty. In the Shuangjing Pilot project, the UTC (as both URO and CRP) evaluated various locations and ultimately selected "Well No.1". The site is a corner space between the community and the road, adjacent to two young and vibrant communities, an art museum, and a railway. It is a low-quality area with great potential for transformation into a vibrant area. The Jiaqichang Community Garden project originated from Tsinghua University's "Qinghe Experiment" project. After repeated inspections of several locations, the SNS finally selected a piece of idle land in the Jiaqichang Community, which is of moderate size and can match the funds provided by the street and the Qinghe experiment project. It can be seen that URCP's measures in the project preparation stage are well-considered. They establish extensive connections with grassroots governments, local companies, or universities to meet budget and manpower needs. The social capital of the URCP mode began to form in the early stages of the project, which to some extent avoided conflicts among complex stakeholders during implementation.

(2)　Stage 2: Product Launches.

Product launch is the core part of the URCP mode. Practice has shown that companies in the URCP have at least one core product in improving the community environment, promoting public interaction, or providing public facilities. They put core products into community space creation and then test and evaluate the effectiveness of the products. In the Shuangjing Pilot project, the UTC used computer recognition algorithms to analyze the movement and behavior of people in the area and then used data visualization software to convert the data into actionable methods (Figure 8). Then, using multi-agent simulation technology, they created initial plans, ran spatial behavior modes, and predicted the results of the transformation [67]. By comparing the advantages and disadvantages of different plans, they further optimized the final plan [68]. In the Jiaqichang Community Garden project, the SNS mainly provides green ecological community construction services. Their designers come from the fields of landscape architecture, planning, and horticulture. After thoroughly investigating the ecological conditions of the site, they formulated ecological restoration and landscape strategies based on the current status of water, soil, and animal and plant resources, and closely monitored the ecological benefits of the site.

(3)　Stage 3: Co-construction within the Social Resource Network.

The construction process of the project can be viewed as the result of the joint action of the physical network and human resource network. In the Jiaqichang Garden case, the SNS adopted a flexible participatory design and construction approach, leading residents, social volunteer groups, and others directly into the site without precise design drawings to construct while designing. This approach effectively supplemented the residents' knowledge and inspired their design inspiration. However, this is inseparable from Tsinghua University's long-term nurturing work and a large number of participatory discussions before the project, such as what functions the garden needs to achieve, what should be placed in different locations, and how to use the least materials to maximize the recycling of materials (Figure 9). While establishing a good co-construction foundation, the team also contacted suburban farms and nurseries and ordered environmentally friendly materials such as soil, wooden stakes, seedlings, and pine bark within the budget. They led

all participants to use these environmentally friendly materials to build garden seats and floors, make insect houses, and plant eco-friendly plants.

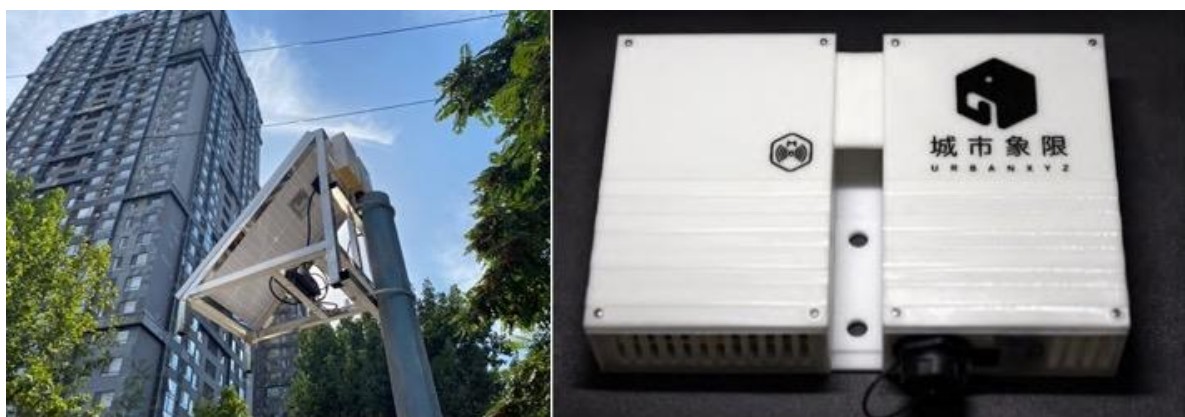

**Figure 8.** Photos of the team's research with sensors. (Figure source: Urbanxyz Tech Company, Beijing, China, 2020).

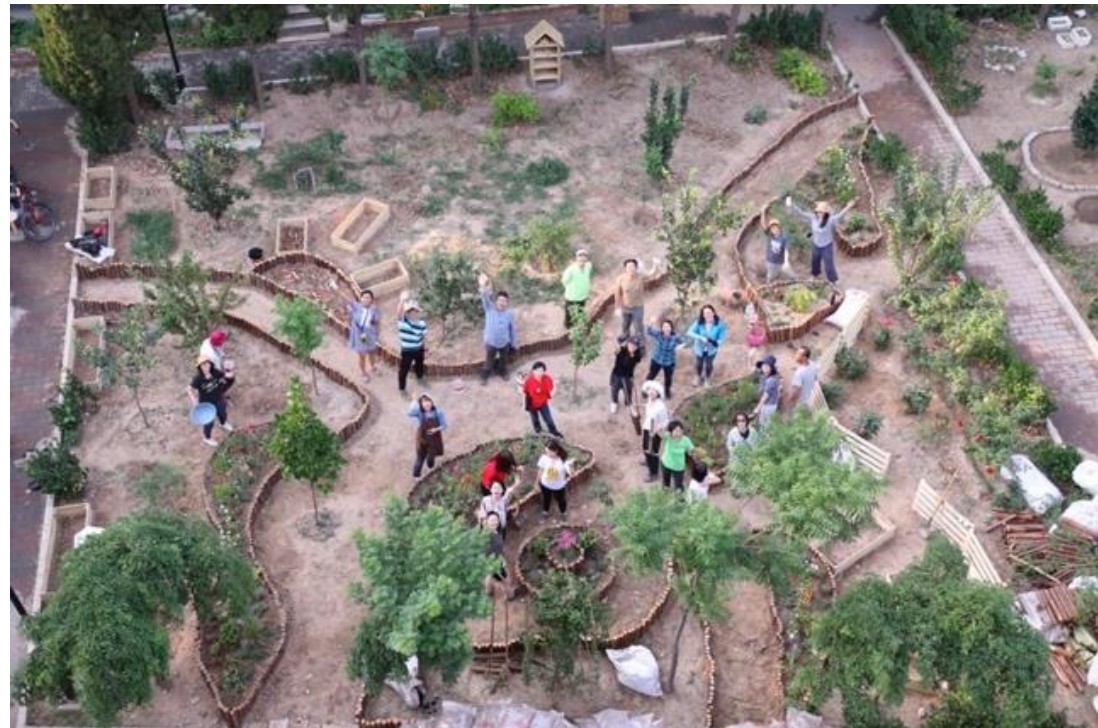

**Figure 9.** Photos of co-construction. (Figure source: Seeding Nature Studio, 2019).

(4)    Stage 4: Maintenance, Operation, and Product Upgrades

In the URCP mode, maintenance and operation can be considered as processes of product upgrade. In addition to the algorithms mentioned earlier, UTC's core products also include smart sensors that monitor environmental quality and human traffic on-site. Once built, the data collected by these sensors helps evaluate the operational status and social value of the pilot project, thereby reducing maintenance costs and expanding its influence. These data are also used to upgrade and supplement existing technologies, including mobile environmental monitoring systems and the Street Brain dashboard (Figure 10). For the Jiaqichang neighborhood, which has mostly elderly and child residents, the SNS has also developed a green education curriculum as an additional product, aiming to teach

residents gardening and maintenance skills while reducing the cost of children's nature education.

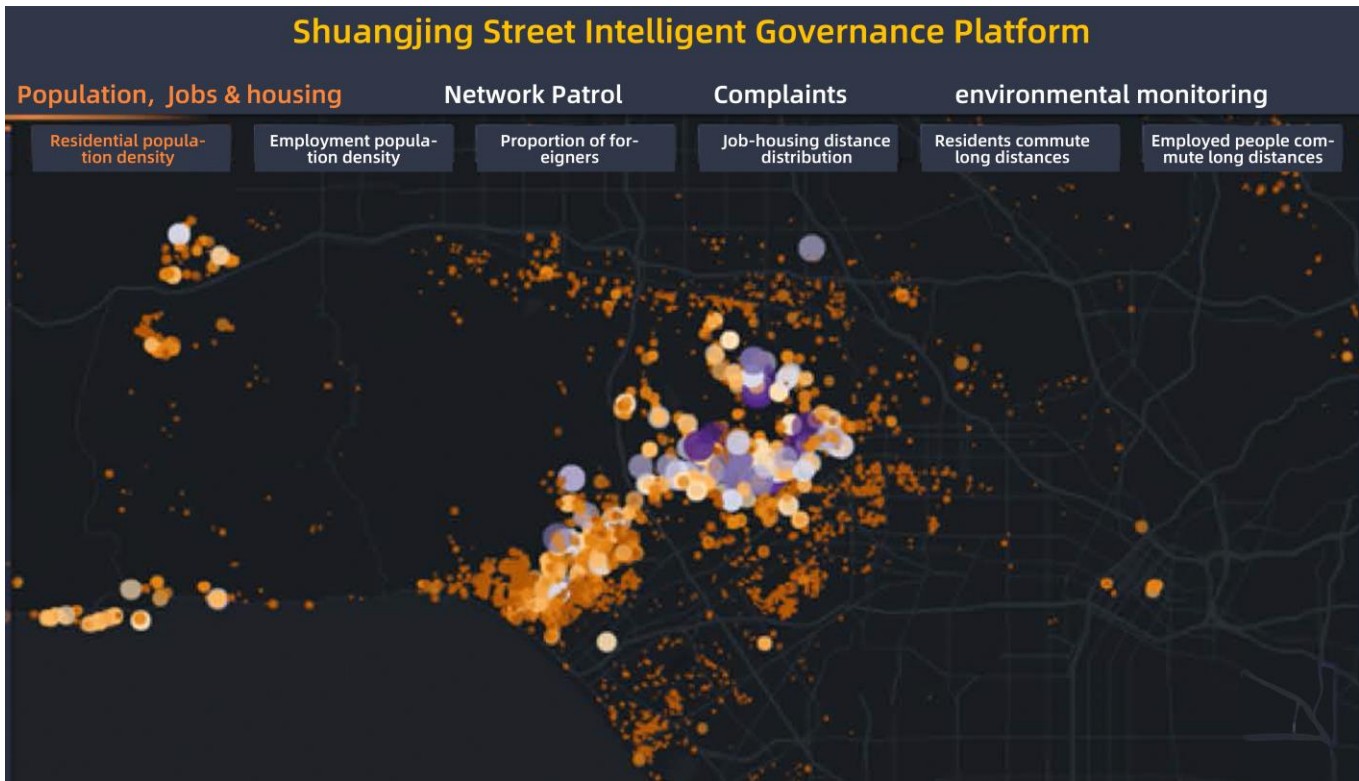

**Figure 10.** Street Brain dashboard. (Figure source: Urbanxyz Tech Company, Beijing, China, 2020).

Table 4 outlines the specific processes in the four stages of the two cases in the URCP model.

3.3.2. The Role of the "Middleman"

In participatory regeneration, the companies of the URCP often play the role of intermediaries (Figure 11). They gather different stakeholders in a common context, reducing communication costs and improving communication efficiency. Firstly, they act as resource coordinators in the government–resident–social cooperation network. They enrich the learning path of community residents based on social resources and promote more innovative experiments for operators and companies to participate in community regeneration. Secondly, they are also the intermediaries for the transmission of information, explaining long-term government policies to the community and providing feedback to the government on issues that need to be addressed in the community.

**Table 4.** Process and participation methods in the URCP mode.

| Stage | Specific Process | Participation Methods | |
|---|---|---|---|
| | | Shuangjing Pilot | Jiaqichang Garden |
| Research and Preparation | Pre-assessment | Fully research the street or community to which you belong and select the most suitable site | |
| | Connecting resources | A cooperative approach led by enterprises and planners, supported by the government, built by residents, participated by society, and supported by public welfare | |

**Table 4.** *Cont.*

| Stage | Specific Process | Participation Methods | |
| --- | --- | --- | --- |
| | | Shuangjing Pilot | Jiaqichang Garden |
| Product Launches | Initial product launch | Smart Tools | Investigation of ecological elements |
| | Information evaluation | Environmental monitoring data, facility usage data, and feedback from residents | Landscape effects and ecological benefits |
| Co-construction within Social Resource Network | Participatory design | Planners guide residents in making designs by explaining the conclusion of data observations | Group discussion on how the space functions and how it is used |
| | Participatory construction | Planting and naming of the site | Residents were involved in the process of soil improvement, planting, and making wooden stake fences and insect houses |
| Maintenance, Operation, and Product Upgrades | Maintenance | Volunteers among residents | Property and smart tools |
| | Sustainable operation | Designers regularly conduct green building classes for residents | Public events, communication sessions |

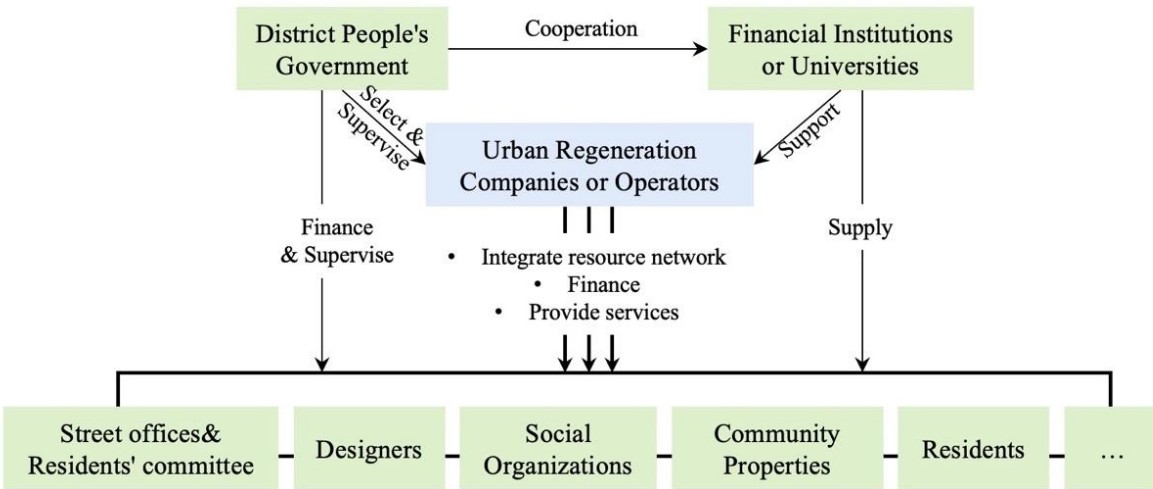

**Figure 11.** Relationship of the URCP mode with stakeholders. (Figure source: made by the author).

## 4. Comparison and Applicable Scenarios of the Three Modes

After the analysis of their features and advantages, the induction of their pathway, and the summary of their relationships with multiple stakeholders, these three modes will be compared comprehensively, especially the similarities and differences in the goals, mechanisms, driving factors, and processes. Based on the similarities and differences, advantages and limitations will be summarized, and their applicable scenarios will be proposed.

### 4.1. Similarities

They are all located in the central urban area of Beijing and were conducted within five years. All six cases and modes share several similarities, such as close connections with the government, similar public participation strategies and update outcomes, and underutilized internal community resources.

### 4.1.1. Close Connections with the Government

While the three modes demonstrate different power distributions between government intervention and public participation, they all have close connections with the government and all exhibit significant differences from the traditional government-led model. The PDG mode is the closest to the government (from the government's initiative to decentralize) and creates the most direct bridge between the residents and the government. Although the PIL mode is initiated by innovation institutions or practitioners, the project's funding generally comes from the street, party, or mass service center to which the community belongs, and the project implementation process is supervised by street personnel. The URCP mode shows similarities more akin to the international model (but the difference is that the URCP is for-profit, mixed-status, and heavily backed by the government). The UTC and SNS, which act as CRPs for Shuangjing Street in Chaoyang District and Qinghe Street in Haidian District, respectively, also have a good basis of cooperation with government agencies such as the Beijing Institute of Urban Planning and Design, the China Academy of Urban Planning and Design, and the Beijing Municipal Bureau of Planning and Natural Resources. Regardless of the model, obtaining endorsement and support from the local government makes it easier to gain residents' trust.

### 4.1.2. Similar Public Participation Strategies and Update Outcomes

All three modes adopt relatively complete public participation processes, allowing residents to participate directly in the decision-making and construction process. However, in reality, the three modes have similarities in the content and format of the public participation process, and there are still phenomena such as narrow coverage and shallow participation. This is because community renewal in China is still in the exploratory stage, and the implementation details of community participation in various regions are not yet complete. Therefore, the strategies and methods used are mainly community meetings, participatory workshops, etc., which do not always effectively attract diverse communities and ensure that more people's needs and preferences are reflected in the decision-making outcomes.

Although all three modes have played a positive role in enhancing functionality and environmental quality, improving neighborly relationships, and promoting social organization construction, these practical effects lack follow-up evaluation and assessment standards. This may be due to the lack of data collection and the absence of a complete feedback mechanism, leaving room for further development in the future. Table 5 demonstrates the actual effects of the three modes in three areas: physical space, community cohesion, and social organization building.

**Table 5.** The effectiveness of the three modes in practice.

|  | The PDG Mode | The PIL Mode | The URCP Mode |
|---|---|---|---|
| Environment improvement | Improve the overall environment of the community | Beautify the environment while highlighting the characteristics | Integrated activity spaces and landscape facilities |
| Community vitality enhancement | Significantly improve community cohesion | Rapidly enhance community vitality on a small scale | Abundant use scenarios to promote interaction |
| Community organizations construction | Form a micro-space governance network | Effectively explore community talent | Increase community capital |

### 4.1.3. Internal Community Resources Need to Be Further Explored

China's community practices involve diverse usage scenarios, requiring in-depth exploration of the characteristics, experiences, and urgent problems to be solved in different communities. International cases have emphasized that active residents have strong adaptability, making them easy to establish trust with community members, cooperate with other stakeholders, and play a mobilizing and supporting role in community regen-

eration [69]. In the practice of the three modes, although actively participating residents can be seen, they are generally doorkeepers or community committee members, with a small number of ordinary residents who are keen to participate in various workshops and seminars, there is still a lack of exploration of other practitioners within the community. These people are likely to also come from the government, the design industry, or technology companies, and they may be the main driving force for future sustainable community construction.

*4.2. Differences*

Despite the above similarities, there are more differences among the three modes, including cooperation mechanisms, targets and drivers, and priorities in the process. It is precisely these differences that reflect the value of different modes.

### 4.2.1. Cooperation Mechanisms

In general, the PDG mode is primarily a top-down pathway with a bottom-up supplement. It relies on the community responsibility planner system and is established by grassroots planners authorized by the government to build bridges between the government, communities, residents, and society, and lead multiple stakeholders in designing and building together. The PIL and URCP modes are both based on cross-departmental cooperation among professionals or teams from different organizations but differ in that the PIL mode is smaller in scale and non-profit and is a form of social service. The URCP mode frequently has a larger scale than the PDG mode, with enterprises and operators typically testing their core products (which could be technology or services) in the field of community regeneration for profit. They are also closely monitored and supported by the government, allowing bottom-up flexibility and top-down standardization to coexist.

### 4.2.2. Targets and Drivers

The PDG mode is driven by the government planning department, which aims to improve the public participation system and explore refined community governance through spatial building. The PIL mode is typically driven by practitioners motivated by personal ideals or social responsibility, with the goal of revitalizing the community and generating positive social benefits. The URCP mode is intended to bring core products into the field of community regeneration and investigate a sustainable commercial operation mode, initially supported by government and corporate funding, and later by the establishment of community capital during the renewal process.

### 4.2.3. Priorities in the Process

The PDG mode's core process is guiding collaboration and reaching consensus. Planners are in charge of building the communication platform as a whole and increasing participation as much as possible, partly to ensure that government policies are fully conveyed, but also to gradually spread awareness of co-construction, co-governance, and sharing through the process of guiding–discussing–feedback–cultivating. Simultaneously, planners in the co-construction process pay close attention to and mobilize the community's weak participants to ensure that more diverse needs and suggestions are heard and adopted. The PIL mode distinguishes itself through customized regeneration planning, which can adjust plans in real-time based on the most recent feedback during the construction process. This implementation path can sufficiently motivate and inspire residents' enthusiasm, ensuring low costs, high participation, and long-term effects. The URCP mode prioritizes product launches. It collects feedback on product usage after it is released in order to improve product quality. Simultaneously, the process of product launch is also a process of establishing local resource networks, which contributes to the discovery of sustainable internal forces within the community. Table 6 illustrates how the three modes differ in terms of objectives, mechanisms, main stakeholders, and working pathways.

**Table 6.** Structures and logic in the three modes.

|  | PDG | PIL | URCP |
|---|---|---|---|
| Targets | Bridging the gap and exploring the institutions | Innovations and good practice | Development and testing of service products |
| Mechanisms | Top-down as dominant, bottom-up as a supplement | Bottom-up as dominant, top-down as a supplement | Top-down with bottom-up |
| Main stakeholders | Professional planners and grassroot state | Individuals, studios, and social entrepreneurs | Corporates, social entrepreneurs, professional planners, grassroot state |
| Working path | Step by step | Spontaneous | Interactive |

*4.3. Advantages and Applicable Scenarios*

It is also because of the different characteristics of the three modes that they are suitable for different scenarios. The strength of the PDG mode is the multi-level governance platform established that effectively communicates the views and demands of various stakeholders. The responsibility of planners, as representatives of government decision-making departments, is to support improving decision-making transparency and residents' trust. However, there is still a need for enhancing the depth of public participation and creating various types of empowerments for residents. Based on the foregoing, the PDG method is appropriate for communities with a big scale, a complicated population composition, or specific social contradictions.

The PIL mode's technological procedures are more refined and efficient when compared to large-scale structures, such as planner teams or operators, allowing practitioners to quickly adjust to ever-changing community needs and assuring maximum satisfaction of bottom-up demands. As for the disadvantages, the impact of the PIL mode is very limited. As a result, the PIL mode is suitable for small-scale projects that must be completed quickly.

For the URCP mode, the strength of several collaborators can transform community resources into usable community capital, lowering project operational costs and encouraging additional social actors, such as businesses, to participate in community regeneration. However, attention needs to be paid to the qualifications of partner companies and the regularity of the implementation process. The URCP mode is better suited to programs that have their own community resources.

**5. Discussion**

Collaborative governance has been studied extensively in large-scale urban regeneration strategies, but grassroots collaborative governance at the community scale has not been studied in depth. The study clarifies the mechanisms, specific pathways, and drivers of collaborative governance theory in very small-scale urban regeneration projects in China. Although China's social organization and participation mechanisms are not yet perfect, the unique institutional context has fostered collaboration with Chinese characteristics. Comparing the similarities and differences of these specific modes helps us to better understand the mechanisms of community regeneration in China, thus providing a valuable exploration of community participation in the East and a valuable reference for other developing countries around the world.

Regarding the first question, based on the case analysis in Beijing, this paper summarizes the three refined modes of collaborative governance in participatory community regeneration practices in the Chinese context: the PDG mode, the PIL mode, and the URCP mode. All three modes demonstrate close ties to the government, but they differ qualitatively from the traditional government-led mode. This is reflected in the fact that the government is gradually empowering various social entities, but this does not mean the full delegation of power. There is always a supervisory role in projects or financial support.

Each of the three modes also has advantages and limitations. The collaborative platform for multi-level governance in the PDG mode allows the views and knowledge of various stakeholders to be effectively conveyed, increasing trust and transparency in the decision-making process. Thus, the PDG mode is suitable for communities with complex social structures, problems, and conflicts. However, conflict in collaborative governance is a direct element that affects its effectiveness [70,71]. Even though the PDG mode shows more government intervention and the conflict between community stakeholders is relatively gentle in both cases (this is partly due to the fact that both cases were cited with a view to avoiding old communities that could generate violent conflict), it does not mean that CRPs can put in less effort in dealing with complex conflicts. Our case studies suggest that platform-based collaborative structures led by CRPs with some official status can be a very effective way of dealing with complex community conflicts but are likely to have consequences, such as rigidity and low effectiveness, in the way the public is engaged.

The PIL mode is characterized by strong innovation, controllable costs, and high efficiency, making it suitable for fast implementation projects that focus primarily on small-scale scene design. However, the influence of the mode is very limited and there is little revenue to the initiators (usually small organizations or individuals). The PIL mode is difficult to sustain if the street or community cannot provide dedicated funding.

The URCP mode involves more stakeholders and is formally more flexible. It is suitable for resource-rich communities and can transform complex resource interactions among many stakeholders into social capital, thereby improving service quality and reducing operating costs. The URCP mode has a moderate level of government intervention and public participation compared to the first two and is generally a mode closer to international experience. However, in our two cases, the initiators themselves both have CRP status, which requires dedicated grassroots administrators to verify the credentials of these alliances and the proportion of their budgets allocated to project implementation and product launch, otherwise there is a risk of credibility issues for grassroots governments. At the same time, the more nuanced power dynamics under multiple identities in the URCP mode deserve further study.

In addition, there are a number of issues that deserve further discussion. As the capital of China, Beijing has demonstrated its innovation and creativity in many practical urban regeneration projects. In fact, there are more than three types of collaborative governance in community regeneration in China. For example, Shanghai has a cooperative group model that is formed spontaneously by community residents out of a sense of hobby and responsibility, and Chongqing has a social capital-leading mode that is responsible for the whole process of construction, maintenance, and management. The emergence of these modes is related to the overall development strategy of the city and the urban context, which can be further summarized in the future.

## 6. Conclusions

With the development of community practices and social innovation in China, participatory regeneration in old communities has received widespread attention, resulting in a variety of distinctive practice pathways. Enterprises, academic institutions, innovative groups, non-profit organizations, and other social participants in major cities are injecting new vitality into the field. The Beijing cases show that most practices in recent years have drawn on international cases in terms of mechanisms, processes, and methods, and all can be seen as applications of collaborative governance theory at the community level.

Our research summarizes three subdivisional modes of collaborative governance and identifies the characteristics of the three modes, the processes, and how stakeholders interact with each other. Our research has important implications for researchers and practitioners in urban regeneration and collaborative governance. Theoretically, we elucidate the mechanisms of collaborative governance for small-scale urban regeneration projects, complementing a more sophisticated model of collaboration in the Chinese context. Practically, we provide clear guidance to grassroots managers, such as how to choose the most

appropriate collaborative model, how to develop the implementation process, which stakeholders should be invited, what roles they play in the collaboration, and what the impacts and negative consequences might be.

**Author Contributions:** Methodology, Y.Q.; Investigation, Z.Z. and J.P.; Resources, Y.Q.; Writing—original draft, Z.Z.; Visualization, Z.Z.; Project administration, Y.Q. All authors have read and agreed to the published version of the manuscript.

**Funding:** This research was funded by Beijing Municipal Social Science Foundation Planning Project, grant number 22SRB010, Natural Science Foundation of Beijing, grant number 9222022.

**Data Availability Statement:** No new data were created or analyzed in this study. Data sharing is not applicable to this article.

**Conflicts of Interest:** The authors declare no conflict of interest.

## Note

1    This data is taken from a preliminary statistic from the Ministry of Housing and Urban Development 2020, the link is: http://m.app.cctv.com/vsetv/detail/C11346/063baba839b547f3bb396b5d3da39b11/index.shtml#0 (accessed on 1 February 2023).

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
