# Peer review of "Collaborative Governance for Participatory Regeneration Practices in Old Residential Communities within the Chinese Context: Cases from Beijing"

_land, doi:10.3390/land12071427_

Round 1

Reviewer 1 Report

Dear Authors, 

The article is interesting and based on good research. The problem under consideration is interesting and I believe that ultimately the article should be published.

However, I am surprised that, having done such good empirical research, the authors have not decided to conduct an in-depth discussion. I believe this text should have at least a 1 page discussion. The Discussion chapter should include a discussion of research results - strengths and weaknesses, and a discussion with research results of other authors on the following issues:

a) participatory regeneration practices in other cities in China and selected in the world

  b) total old residential communities

c) issues of social development in Beijing

Other

Conclusions chapter is too long. The first half of this chapter is well element to use in Discussion. 

Fig. should also be enlarged. 1.

Sincerely

Author Response

回复审稿人 1 条评论

第 1 点:我认为这篇文章应该至少有 1 页的讨论。讨论章节应包括对研究结果的讨论——优点和缺点,以及与其他作者关于以下问题的研究结果的讨论:

  1. a) 中国其他城市以及世界各地的参与式再生实践
  2. b) 老旧住宅小区总数
  3. c) 北京社会发展问题

回复1:非常感谢您对本文研究设计提出的建议。我修改了讨论部分的结构,添加了三种模型可能产生的负面后果。

  1. 本文仅考察中国背景下的再生实践,因此在讨论部分添加了上海和重庆的案例,以表明本文讨论的三种模式并不是实践中唯一存在的模式。
  2. 引言中提到了老旧小区总数,数据来自住建部2020年初步统计,目前尚无更精细的数据。
  3. 北京的社会发展问题不是本文讨论的重要内容。作为中国具有代表性的特大城市之一,北京的六个案例的模式在其他城市也有类似的表现,不同之处可能在于每个城市都有自己的社区规划师体系,因此社区规划师负责的具体任务也不同。实际项目中并不相同。

第2点:结论章节太长。本章的前半部分非常适合在讨论中使用。图也应该放大

回复2:根据您的建议,结论已缩短,图片尺寸已放大。

Reviewer 2 Report

This paper is overall well written and contain a good overview of the participatory regeneration practices as currently undertaken in China.

Both literature review and presentation of the methodology are exhaustive and comprehensive, with significant discussion and conclusions.

Some editing is required before it can be accepted for publication:

1- Add some reference with respect to the statement at p1. lines 31-31(As of 2020, there were about 1.7 million old...)

2-At p.4 line 135; 137; 138; 152; 160; 167; 173 and elsewhere in other pages there is reference to: Z Foundation; X design; C Community, B University, U Technology Company, etc. Authors should indicate what these capital letters means; if the names of the various entities cannot be disclosed for any reason, it should be then clearly mentioned in a separate note.

3-Interlines between page 1-3 and pages 3-25 are different; and there are a few typos at References [2] p. 22 concerning the separation of letters for the word "residential" and "renovation".

4- Table 1 at page 5 should include an additional column indicating the case study (e.g. Case Study A) and the name should be both in English and Chinese characters.

5- All figures in the paper should mention the source and the year.

Author Response

Response to Reviewer 2 Comments

Point 1: Add some reference with respect to the statement at p1. lines 31-31(As of 2020, there were about 1.7 million old...)

Response 1: Thank you for your very careful review. Sources of data have been indicated.

Point 2: At p.4 line 135; 137; 138; 152; 160; 167; 173 and elsewhere in other pages there is reference to: Z Foundation; X design; C Community, B University, U Technology Company, etc. Authors should indicate what these capital letters means; if the names of the various entities cannot be disclosed for any reason, it should be then clearly mentioned in a separate note.

Response 2: All abbreviations have been specified throughout and are used in parentheses and later in the text.

Point 3: Interlines between page 1-3 and pages 3-25 are different; and there are a few typos at References [2] p. 22 concerning the separation of letters for the word "residential" and "renovation".

Response 3: Spelling errors have been corrected.

Point 4: Table 1 at page 5 should include an additional column indicating the case study (e.g. Case Study A) and the name should be both in English and Chinese characters.

Response 4: Tables have been amended.

Point 5: All figures in the paper should mention the source and the year.

Response 5: Image sources have been cited.

Reviewer 3 Report

Overall, the article "Participatory Regeneration Practices in Old Residential Communities within Chinese Context: Cases from Beijing" provides valuable insights into the implementation processes and driving mechanisms of participatory community regeneration pathways in six cases in Beijing. However, there are several areas where the article could be improved.

Firstly, the methodology used in the study is not clearly described. It is unclear how the six cases were selected and categorized into three modes, and whether any criteria were used to determine which mode each case belonged to. Additionally, the article does not provide information on the sample size or the data collection methods used in the study.

Secondly, while the article provides a detailed description of the Jiaqichang community garden project, it is unclear how this case study was chosen and why it was singled out for a more in-depth analysis. The article would benefit from a more detailed explanation of the selection process for this case study.

Thirdly, the article lacks a clear theoretical framework to guide the analysis. While the three modes identified in the study are useful for organizing the cases, there is no discussion of any theoretical perspectives that could help to explain why these modes have emerged in the Chinese context or how they relate to broader debates in the field of participatory community regeneration.

Finally, the article could benefit from a more critical approach to the analysis of the cases. While the article notes some limitations of the different modes, it does not engage with any potential drawbacks or negative consequences of these approaches. A more nuanced analysis that considers both the benefits and limitations of each mode would provide a more comprehensive understanding of the different pathways of participatory community regeneration in the Chinese context.

In summary, while the article provides valuable insights into participatory community regeneration practices in Beijing, there are several areas where it could be improved, including a clearer methodology, a more detailed explanation of the case study selection process, a stronger theoretical framework, and a more critical analysis of the different modes identified in the study.

Author Response

Response to Reviewer 3 Comments

Point 1: Firstly, the methodology used in the study is not clearly described. It is unclear how the six cases were selected and categorized into three modes, and whether any criteria were used to determine which mode each case belonged to. Additionally, the article does not provide information on the sample size or the data collection methods used in the study.

Response 1: The methods used in the study were mainly collected and interpreted within the framework of rooted theory. Information on the cases was obtained from fieldwork, official documents and semi-structured interviews in field research.

The six cases are all representative and consistently influential cases in the central city of Beijing, and we believe they are representative of the prevailing paths of participatory community regeneration today. After sorting through the context, initiators and implementation processes of the six cases, we have identified three identifiable mainstream approaches - they are all essentially explorations of innovative collaborative governance, but the identity of the initiators, the specific ways of collaboration and the power dynamics between stakeholders are the fundamental elements that distinguish the three pathways.

Point 2: Secondly, while the article provides a detailed description of the Jiaqichang community garden project, it is unclear how this case study was chosen and why it was singled out for a more in-depth analysis. The article would benefit from a more detailed explanation of the selection process for this case study.

Response 2: Thank you for your very careful review. This article does not provide a separate detailed description of the Jiaqichang Gardens case. Each section in Chapter 3 of this study summarises the characteristics of the three models and uses the two corresponding cases of each as arguments.

Jiaqichang Garden impresses you, probably because it belongs to the more complex URCP model, and is therefore explained at a slightly longer length.

Point 3: Thirdly, the article lacks a clear theoretical framework to guide the analysis. While the three modes identified in the study are useful for organizing the cases, there is no discussion of any theoretical perspectives that could help to explain why these modes have emerged in the Chinese context or how they relate to broader debates in the field of participatory community regeneration.

Response 3: Based on your suggestions, we have added the theory of cooperative governance as a framework. All three models can be seen as applications of collaborative governance theory to community level renewal projects.

Point 4: Finally, the article could benefit from a more critical approach to the analysis of the cases. While the article notes some limitations of the different modes, it does not engage with any potential drawbacks or negative consequences of these approaches. A more nuanced analysis that considers both the benefits and limitations of each mode would provide a more comprehensive understanding of the different pathways of participatory community regeneration in the Chinese context.

Response 4: The discussion and conclusion sections of this paper have been revised. The potential negative consequences of the three models have been added to the discussion to help the reader understand more fully the pathways of collaborative governance in participatory community renewal.

Round 2

Reviewer 1 Report

Dear Authors

I have on more remarks,

Sincerely

Reviewer 3 Report

All comments have been answered, and we see that the research has become suitable for publication.